# Five Days of Tart Cherry Supplementation Improves Exercise Performance in Normobaric Hypoxia

**DOI:** 10.3390/nu15020388

**Published:** 2023-01-12

**Authors:** Masahiro Horiuchi, Yoshiyuki Fukuoka, Katsuhiro Koyama, Samuel J. Oliver

**Affiliations:** 1Faculty of Sports and Life Science, National Institute of Fitness and Sports in KANOYA, Kagoshima 8912393, Japan; 2Division of Human Environmental Science, Mt. Fuji Research Institute, Yamanashi 4030005, Japan; 3Faculty of Health and Sports Science, Doshisha University, Kyoto 6100394, Japan; 4Faculty of Sport Science, Yamanashi Gakuin University, Yamanashi 4008575, Japan; 5Institute for Applied Human Physiology, School of Human and Behavioural Sciences, College of Human Sciences, Bangor University, Bangor LL57 2DG, UK

**Keywords:** antioxidant, blood flow, DNA damage, O_2_ extraction, oxidative stress, tissue oxygenation, vasodilation

## Abstract

Previous studies have shown tart cherry (TC) to improve exercise performance in normoxia. The effect of TC on hypoxic exercise performance is unknown. This study investigated the effects of 5 days of tart cherry (TC) or placebo (PL) supplementation on hypoxic exercise performance. Thirteen healthy participants completed an incremental cycle exercise test to exhaustion (TTE) under two conditions: (i) hypoxia (13% O_2_) with PL and (ii) hypoxia with TC (200 mg anthocyanin per day for 4 days and 100 mg on day 5). Pulmonary gas exchange variables, peripheral arterial oxygen saturation (SpO_2_), deoxygenated hemoglobin (HHb), and tissue oxygen saturation (StO_2_) assessed by near-infrared spectroscopy in the vastus lateralis muscle were measured at rest and during exercise. Urinary 8-hydro-2′ deoxyguanosine (8-OHdG) excretion was evaluated pre-exercise and 1 and 5 h post-exercise. The TTE after TC (940 ± 84 s, mean ± standard deviation) was longer than after PL (912 ± 63 s, *p* < 0.05). During submaximal hypoxic exercise, HHb was lower and StO_2_ and SpO_2_ were higher after TC than PL. Moreover, a significant interaction (supplements × time) in urinary 8-OHdG excretion was found (*p* < 0.05), whereby 1 h post-exercise increases in urinary 8-OHdG excretion tended to be attenuated after TC. These findings indicate that short-term dietary TC supplementation improved hypoxic exercise tolerance, perhaps due to lower HHb and higher StO_2_ in the working muscles during submaximal exercise.

## 1. Introduction

Improvements in endurance exercise performance, such as time trials under a given distance or time to exhaustion, are important for endurance athletes. It has been reported that tart cherry (TC) might improve endurance exercise performance via the facilitation of oxygen delivery ($\overset{.}{Q}$O_2_) to exercising muscles [1], perhaps due to possible mechanisms by increasing nitric oxide (NO) bioavailability [2]. A recent meta-analysis summarized that TC supplementation significantly improved exercise performance (standardized mean difference: 0.36; 95% confidential interval: 0.07 to 0.64, *p* = 0.01) upon pooling previous literature [3]. However, when viewed separately, some studies have shown that dietary TC supplementation extended the time to exhaustion or shortened time trials [1,4]; however, others had contradictory findings [5,6]. One possible explanation for these discrepancies may be related to different study settings (e.g., dosage of supplement, exercise mode, or population features). It should be noted that these studies were conducted only under normoxia [1,4,5,6]. Another distinctive advantage of TC supplementation has been proposed to reduce subset of oxidative stress markers [7,8]. However, whether this property would improve exercise performance seems to be uncertain. 

Hypoxic exposure elevates several biomarkers of oxidative stress such as 8-hydro-2′ deoxyguanosine (8-OHdG) [9,10,11,12,13] or 2-thiobarbituric acid-reactive substances [14,15,16] as indexes of oxidative damage to DNA or lipids, as well as exercise-induced elevated oxidative stress [17]. In general, increased oxidative stress may be associated with impaired vascular function, resulting in insufficient oxygen delivery ($\overset{.}{Q}$O_2_) to exercising muscles during exercise with the following mechanisms: (i) reducing the bioavailability of NO [18], (ii) increasing sympathetically mediated vascular tone [19], (iii) enhancing angiotensin II-mediated vasoconstriction [20], and (iv) endothelin-1 activity [18,21].

In this regard, TC could potentially improve hypoxic exercise performance by increasing the $\overset{.}{Q}$O_2_ to exercising muscles [1] via increasing NO bioavailability [2] and hypoxia-induced vasodilation during exercise [22]. These effects would (presumably) serve to improve $\overset{.}{Q}$O_2_-to-O_2_ uptake ($\overset{.}{V}$O_2_) matching, thereby increasing muscle and microvascular O_2_ pressure (PO_2_) and enhancing blood myocyte O_2_ flux and mitochondrial control by increasing intracellular PO_2_ [23,24]. TC supplementation may also improve exercise performance via an antioxidant effect that prolongs the optimal cellular redox state for force production [25]. 

Accordingly, it was hypothesized that TC supplementation would improve exercise performance under hypoxia, lower oxidative stress, and increase tissue oxygenation in exercising muscles compared to placebo. Tissue oxygenation and deoxygenation kinetics in exercising vastus lateralis muscles were evaluated by near-infrared spectroscopy (NIRS). 

## 2. Materials and Methods

### 2.1. The Sample Size and Participants

A previous study reported that 7 days of TC supplementation improved time trial performance by ~4.7% (effect size = 0.78) [1]. To estimate the required sample size, one- and two-tailed paired t-tests were performed with an error probability of 0.05 (α), a power (1-β) of > 0.80, and an effect size of 0.78 using G Power 3.1 analysis software [26]. The required sample size of 12 (one-tailed) or 15 (two-tailed) participants was estimated.

Fifteen young, healthy, recreationally active volunteers (11 men and 4 women) were enrolled using a digital and paper flyer in the local community (Kawaguchiko town and Fuji-yoshida city, Yamanashi, Japan) and an adjacent university (Health Science University). Exclusion criteria included pregnancy, stationary bike use, current smoker, any known cardiometabolic disorders, use of medications known to affect cardiovascular responses, and regular engagement in moderate-to-vigorous physical activity (≥120 min/week) [27]. The participants were stratified by sex and randomly assigned to the starting condition. Women were studied during the follicular phase just after menstruation based on their basal body temperature and self-report [27]. After a detailed explanation of the study procedure and the possible risks and benefits of participation in this study, each participant signed an informed consent form. This study was approved by the Ethical Committee of Mount Fuji Research Institute in Japan and was performed in accordance with the guidelines of the Declaration of Helsinki (No. 202001).

### 2.2. Experimental Procedures

As shown in Figure 1, this study was a randomized, double-blinded, placebo-controlled crossover design with two experimental conditions including incremental normobaric hypoxic exercise (fraction of inspired oxygen [FiO_2_] = 0.13, equivalent altitude is approximately 3800 m) (i) with a placebo supplementation (PL) and (ii) with TC supplementation. The participants were requested to abstain from caffeinated beverages for 12 h and strenuous exercise and alcohol for 24 h before each session. The participants visited the laboratory on three occasions, including one familiarization with all measurement techniques (i.e., wearing a mask, hypoxic gas inspiration, and semi-recumbent leg cycling exercise) and two experimental visits. During the second and third visits, the participants were randomized to PL or TC. The order of the trials was counterbalanced. These two trials were performed at the same time (08:30–11:30 h) to avoid the effect of a circadian rhythm with at least a 10-day wash-out period. All studies were performed in an environmental chamber (TBR-4, 5SA2GX, Tabai Espec Co., Ltd., Tokyo, Japan) set at a room temperature of 24 °C and relative humidity of 50%. After a 15-minute semi-recumbent position during the resting period, all participants performed an incremental maximal leg cycling test starting for 4 min at a power output for men and women of 40 W or 30 W, respectively (stage 1), followed by 4 min at 80 W or 60 W (stage 2) and 4 min at 120 W or 90 W (stage 3). Subsequently, the power output increased by 20 W or 10 W per min until exhaustion for men or women, respectively (Figure 1). The participants pedaled at a cadence of 60 rpm as set by a metronome. The criteria for exhaustion were as follows: (1) no increase in $\overset{.}{V}$O_2_ despite a further increase in work rate, (2) heart rate (HR) at 90% of the age-predicted maximal value (208 − age × 0.7), (3) a rating of 19 on Borg’s scale of perceived exertion, or (4) can no longer maintain the pedaling rate above 50 rpm despite strong verbal encouragement. The test was terminated when participants met at least one of these 3 criteria and could not maintain the pedaling rate of 50 rpm despite strong verbal encouragement [28].

### 2.3. Supplementation Protocol

Participants were assigned in a double-blinded and randomized manner to ingest TC (tart cherry 1200 mg capsule containing 100 mg of anthocyanin, Nature’s Life, Orem, UT, USA) or a flour placebo. The TC and PL supplementations were visually indistinguishable as they were ground to powder and encapsulated in a gelatin capsule. Participants were instructed to ingest one capsule at 08:00 h and one capsule at 18:00 h for 4 consecutive days before the main experiment, and one capsule 2 h before exercise on the day of the main experiment. The selected TC dose was based on a recent meta-analysis [3] where daily anthocyanin supplementation ranged from 40 to 270 mg per day. Additionally, participants were provided with a list of foods rich in antioxidants and instructed to avoid the consumption of these foods and maintain their normal dietary intake for the duration of the study. 

### 2.4. Measurements

#### 2.4.1. Cardiorespiratory Variables

Pulmonary ventilation and gas exchange variables were measured using a breath-by-breath metabolic measurement system (AE-310S; Minato Medical Science, Osaka, Japan). The inspired and expired gas volumes were measured using a hot-wire respiratory flow system. Flow signals were electrically integrated for the duration of each breath to calculate minute ventilation. The expired fractions of O_2_ and CO_2_ were analyzed using an O_2_ and CO_2_ gas analyzer. Standard gases (O_2_ 15.23%, CO_2_ 4.999%, and N_2_ balance) and room air were used to calibrate the gas analyzer before each test. HR and peripheral arterial oxygen saturation (SpO_2_) were continuously measured with a wireless heart rate monitor (Polar RC800X; Polar Electro Japan, Tokyo, Japan) and a pulse oximeter (WB-100; Nihon Seimitsu Sokki Co., Ltd., Gunma, Japan). 

#### 2.4.2. Tissue Oxygenation Profiles

Local muscle oxygenation profiles at the vastus lateralis muscle (active muscle) were measured using NIRS (BOM-L1TRW; Omegawave, Tokyo, Japan), as previously described [24]. This device uses three laser diodes (780, 810, and 830 nm) and calculates the relative tissue levels of oxygenated and deoxygenated hemoglobin (HbO_2_ and HHb) according to the modified Beer–Lambert law [29]. Tissue O_2_ saturation (StO_2_) was calculated by dividing HbO_2_ by total Hb (HbO_2_ plus HHb). NIRS optodes were placed on the lower third of the vastus lateralis muscle (10–12 cm above the knee joint) [30]. The measurement depth of the NIRS signal was approximately half the distance between the two fiber optic bundles placed over the skin, one comprising the light source and detector [31]. With this in mind, we used 40 mm between the probes, which provides a NIRS signal traverse distance of ~20 mm. This would have allowed the appropriate depth to sample from the vastus lateralis muscles, because the sum of adipose tissue and muscle thickness in vastus lateralis muscle was >20 mm [31]. Indeed, the measured adipose tissue and muscle thickness were 5.4 ± 1.1 mm and 20.6 ± 2.6 mm. These numbers indicate that the near-infrared light was transmitted to the desired muscle bed. The probe holder contained one light source probe, and two detectors were placed 20 mm (detector 1) and 40 mm (detector 2) away from the source. The Hb concentrations received by detector 1 were subtracted from those received by detector 2. This procedure allowed us to minimize the influence of skin blood flow [29].

#### 2.4.3. Urine Sample and Analysis

Total urine samples were collected in a light-shielding bottle (500 mL) pre-exercise and 1 and 5 h post-exercise, and immediately stored at −80  °C for further analysis. At each sampling point, urine volume was measured, and the collection time was recorded. We also recorded the time of urination just before exercise. Urinary 8-OHdG, an oxidative derivative of deoxyguanosine implying oxidative damage to DNA [32], was analyzed. 

### 2.5. Data Analysis

At rest, all cardiorespiratory variables and NIRS signals were averaged over the last 3 min immediately before the exercise (resting baseline values). During submaximal exercise for 4 min at each exercise intensity (i.e., 40–80–120 W and 30–60–90 W for men and women, respectively), these physiological values were averaged over the last 1 min at each exercise intensity. At the maximal level, data were averaged over the last 30 s prior to exhaustion. As our NIRS device represents HHb and total Hb signals as arbitrary units, to compare both signals between participants, the changes in HHb and total Hb were quantified as percentages from the resting baseline values. Specifically, resting baseline values in each trial were defined as 100% and were shown as relative changes. Based on a previous study [33], urinary 8-OHdG excretion was calculated and represented per individual body weight per given time, i.e., as the unit of ng·kg^−1^·h^−1^.

### 2.6. Statistical Analyses

All values were represented as mean ± standard deviation (SD). Statistical analyses were performed using the commercial software Jamovi ver. 2.2.5. Only participants with complete data for primary outcomes were included in the analyses. A paired t-test was used to compare the time to exhaustion between PL and TC. Two-way (trial × time points) repeated measures analysis of variance (ANOVA) was performed to compare all physiological variables. When *p* values were <0.05, Tukey’s post hoc test was used for further comparisons. Effect sizes were calculated as Cohen’s d, defined as small (0.2), medium (0.5), and large (0.8) for paired *t*-tests, and as η^2^, defined as small (0.01), medium (0.06), and large (0.14) effects [34]. A *p* value < 0.05 was considered statistically significant. 

## 3. Results

Two of the fifteen participants claimed severe headache and nausea during hypoxic exercise and had to drop out of the experiments (one participant claimed under PL condition, and the other claimed under TC condition). Thus, 13 participants (10 men and 3 women) aged 21 ± 1 years, with a body mass index of 22.1 ± 2.7 kg·(m^2^)^−1^, completed all trials and were used for further analysis.

### 3.1. Exercise Performance in Normobaric Hypoxia

TC supplementation improved the time to exhaustion compared with that of PL (940 ± 84 s with TC vs. 912 ± 63 s with PL, t = 2.87, Cohen’s d = 0.80, *p* = 0.01) (Figure 2). Specifically, 10 of 13 participants reported improved exercise performance after TC.

### 3.2. Cardiorespiratory Responses at Rest and During Exercise

A two-way repeated measures ANOVA revealed no interaction effects in any cardiorespiratory variables ($\overset{.}{V}$O_2_, carbon dioxide output ($\overset{.}{V}$CO_2_), respiratory gas exchange ratio (RER), pulmonary ventilation ($\overset{.}{V}$_E_), HR, and SpO_2_) (all *p* > 0.05). Further, no condition effects were observed for $\overset{.}{V}$O_2_, $\overset{.}{V}$CO_2_, RER, $\overset{.}{V}$_E_, and HR (all *p* > 0.05) (Table 1). In contrast, there was a significant condition effect for SpO_2_ (*p* = 0.001), where SpO_2_ was higher after TC than PL. As expected, there were also significant main effects of time for all cardiorespiratory variables (all *p* < 0.001).

### 3.3. Muscle Oxygenation Profiles during Exercise

Figure 3 shows a typical example (panel A) and the averaged values of % changes (panel B) in the HHb. During exercise, HHb values at Stages 1 (40 or 30 W for men or women), 2 (80 or 60 W), and 3 (120 or 90 W) were lower with TC than PL (Figure 3A). A typical example (panel A) and averaged values of % changed (panel B) in StO_2_ are shown in Figure 4. During exercise, StO_2_ values at Stages 1, 2, and 3 were higher with TC than PL (Figure 4A,B). Total Hb gradually increased (F = 39.01, η^2^ = 0.228, *p* < 0.001) during exercise with no condition (F = 0.008, η^2^ = 0.000, *p* = 0.93) and interaction effects (F = 2.06, η^2^ = 0.007, *p* = 0.12) (Figure 5).

### 3.4. Oxidative Stress Marker in Urine

Figure 6A shows comparisons in urinary 8-OHdG excretion at rest and 1 and 5 h post-exercise after TC and PL. Significant time (F = 30.06, η^2^ = 0.413, *p* < 0.001) and interaction (F = 5.375, η^2^ = 0.028, *p* = 0.01) effects with no condition effect (F = 0.218, η^2^ = 0.001, *p* = 0.65) were found. Tukey post hoc tests revealed that 1 h post-exercise 8-OHdG was increased from rest after PL and TC (Figure 6A, *p* < 0.05), with the increase tending to be greater after PL than TC (Figure 6B, *p* = 0.07). 

## 4. Discussion

This is the first study to investigate the effects of dietary TC supplementation on exercise performance under hypoxia. The primary findings of the present study were three-fold; compared with PL condition, five days of dietary TC supplementation (i) extended time to exhaustion during an incremental exercise test, (ii) lowered HHb and increased StO_2_ in the vastus lateralis muscle, and increased SpO_2_ during hypoxic exercise, and (iii) attenuated urinary 8-OHdG excretion 1 h post-exercise, which suggests reduced oxidative stress. 

### 4.1. Exercise Performance in Normobaric Hypoxia

Previous studies have shown that TC supplementation improves exercise performance in normoxia [1,4]. The current study extends these findings by showing for the first time that TC supplementation can also improve exercise performance in a hypoxic environment. Comparing the effect sizes of this (d = 0.80) and a previous study (d = 0.78) [1] highlights that the improvement in exercise performance after TC supplementation is similar irrespective of different inspired oxygen concentrations.

### 4.2. Systemic Arterial and Local Muscle O_2_ Saturation 

In the present study, TC supplementation decreased HHb and increased StO_2_ during incremental exercise (Figure 3 and Figure 4) in hypoxia, which provides one explanation for the improved exercise performance after TC supplementation. Lower HHb suggests enhancements in $\overset{.}{Q}$O_2_-to-$\overset{.}{V}$O_2_ matching, resulting in lower fractional O_2_ extraction during incremental exercise to meet muscle demands, especially in the vastus lateralis muscle [35,36], perhaps due to greater venous O_2_ content and higher arterial O_2_ content (SpO_2_ as an index of arterial O_2_ content) was observed with TC (Table 1).

Muscle StO_2_ is calculated by dividing HbO_2_ by total Hb (HbO_2_ plus HHb). In the present study, total Hb gradually increased with no differences between the conditions (Figure 5), which was associated with no differences in microvascular Hb volume [37,38,39]. Therefore, higher StO_2_ with TC might be caused either by higher HbO_2_ and/or lower HHb. Although HbO_2_ may be affected by cutaneous blood flow [40], and thus, we cannot completely exclude HbO_2_ effects on cutaneous circulation. However, lower HHb found after TC supplementation (Figure 3) indicates sufficient $\overset{.}{Q}$O_2_ and TC-induced vasodilation in the vastus lateralis (exercising) muscle, which may consequently have led to higher StO_2_ and SpO_2_, and more efficient metabolism during exercise. This explanation has been suggested in a previous study [41] using similar vasodilatory supplements. 

There are several possible explanations for these findings. It has been suggested that TC supplementation induces vasodilation via NO bioavailability [2]. Previous studies reported that antioxidant supplementation with similar supplements as used in the present study augmented endothelium-dependent vasodilation [42,43,44] through NO-dependent mechanisms [43]. However, this study was conducted using a local muscle exercise (handgrip) for older adults [43]. Therefore, further investigation is warranted to confirm this underlying mechanism in exercise involving large muscle groups as in the present study. 

Some possibilities can be included. Blockade of NO synthase increased muscle O_2_ extraction during handgrip exercise under hypoxia (SpO_2_ ∼85%) [45] and a-vO_2_ difference during knee extension exercise under normoxia [46], reflecting increases in HHb. Additionally, blockade of NO synthase during knee extension exercise under normoxia decreased venous oxygen saturation [47], which may lead to lower StO_2_. Taken together, TC supplementation, which is known to have antioxidant-induced vasodilation effects [43], altered systemic and local oxygen saturation during hypoxic exercise. In another respect, dietary nitrate (NO_3_) supplementation reduced HHb and increased StO_2_, which potentially caused vasodilation via the NO_3_^−^–nitrite (NO_2_^−^)–NO pathway [48] and improved hypoxic exercise performance [29,49]. These results are in line with our findings and support our hypothesis (Figure 2). 

### 4.3. Effects of Tart Cherry Supplementation on Oxidative Stress

The present study identified an attenuated increase in the oxidative stress marker urinary 8-OHdG excretion. Indeed, 1 h post-exercise, the increase from rest in urinary 8-OHdG was halved after TC compared to PL (160% PL vs. and 75% TC Figure 6). These findings agree with previous studies that have also reported that TC supplementation attenuated an increase in lipid peroxide as an oxidative stress marker [5,7] and increased total antioxidant status [5]. Although the oxidative stress markers were different between our study and previous studies [5,7], our findings might be supported by these studies [5,7]. The underlying mechanisms that account for the present results may be related to NO bioavailability. Recent studies demonstrated that black soybean, which includes rich polyphenols with antioxidant effects, improved vascular stiffness, increased urinary NO_2_ and NO_3_ levels, and decreased urinary 8-OHdG in healthy adults [50,51]. Moreover, it has been suggested that anthocyanins activate Nrf-2, which increases expressions of detoxifying enzymes and antioxidant enzymes, resulting in the elimination of reactive oxygen species and oxidant-induced injury cells [52,53]. Our study design cannot clarify these possible mechanisms, and hence, future studies are needed. 

### 4.4. Methodological Considerations and Potential Implications

Several limitations should be considered when interpreting our findings. First, blood flow or vascular conductance as indexes of vasodilation in the working muscles were not evaluated. However, it may be difficult to precisely and continuously measure blood flow in the working muscle during leg cycling (e.g., at the femoral artery). Further, our exercise mode (leg cycling) may be more practically relevant than a local exercise, such as a handgrip or knee extension exercise, which has been conducted with the measurement of blood flow in previous studies [43,54]. Second, the present study did not conduct a normoxic exercise with or without TC supplementation as a control condition. However, our main aim was to investigate the effects of dietary TC supplementation on hypoxic exercise performance and not to compare normoxic and hypoxic exercise with TC. Third, rather than assessing a broad range of oxidative stress markers, only one oxidative stress marker, urinary 8-OHdG excretion was evaluated. Nonetheless, the urinary excretion of 8-OHdG has been recognized as a stable biomarker of DNA oxidative damage and reflects overall systemic oxidative stress level [55]. Fourth, to date, studies examining of TC supplementation effects on exercise have used a variety of supplements produced using different production methods [3], and hence, it is difficult to draw direct comparisons between these studies and determine a consensus for an appropriate dose for sports performance. However, our results demonstrate that whole-body exercise tolerance in hypoxia is improved with 5 days of TC supplementation in healthy young men and women. As a practical implication, examples include mountain stages in cycling (up to 2800 m in the Tour de France) and mountain running events (e.g., Mount Fuji, up to 3776 m in Japan, or Pikes Peak, up to 4300 m in Colorado, USA). Thus, our findings can potentially be applied to these elite sports. 

## 5. Conclusions

Short-term dietary TC supplementation improved hypoxic exercise performance. The improved exercise performance may be explained by decreases in HHb levels and increases in StO_2_ levels in the working (vastus lateralis) muscles during exercise. Moreover, TC supplementation may attenuate oxidative stress, as indicated by attenuated urinary 8-OHdG excretion rates 1 h post-exercise.

## Figures and Tables

**Figure 1 nutrients-15-00388-f001:**
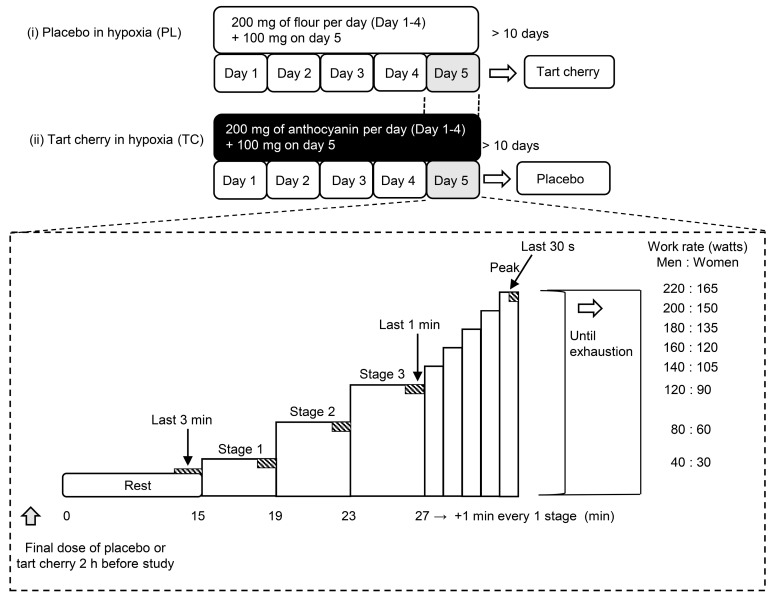
Study protocol of the present study. PC; placebo, TC; tart cherry. Rectangles with diagonal lines indicate the cardiorespiratory and near-infrared spectroscopy measurement period.

**Figure 2 nutrients-15-00388-f002:**
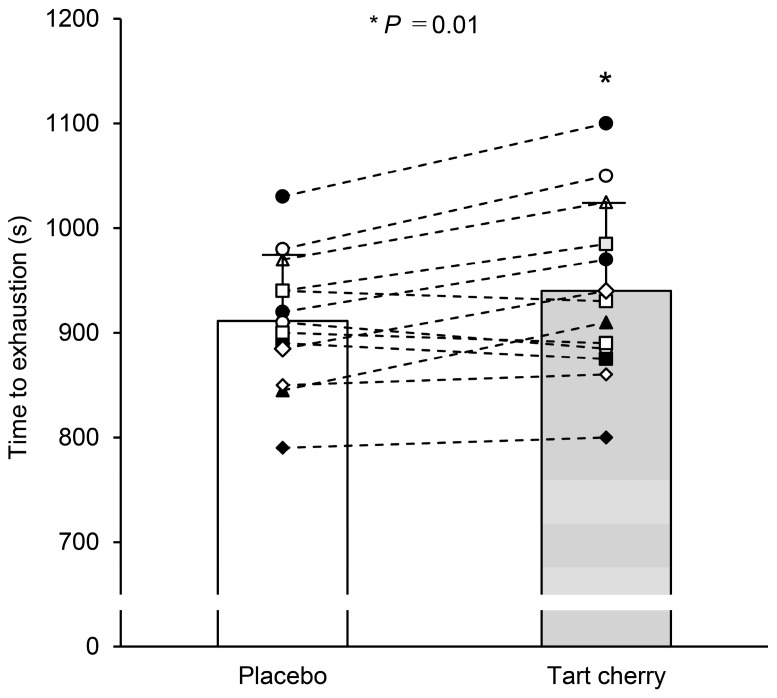
Time to exhaustion during incremental hypoxic exercise after 5 days of placebo and tart cherry supplementation. Different symbols with dotted lines indicate individual data, and the bars indicate supplementation group mean values. Values are the mean ± standard deviation (SD). * *p* < 0.05 between PL and TC.

**Figure 3 nutrients-15-00388-f003:**
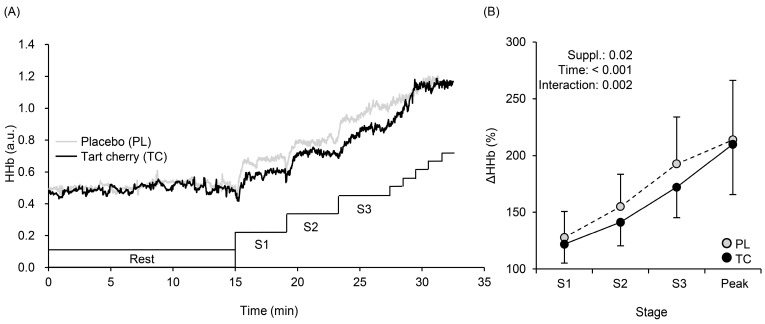
The typical time course changes in muscle deoxygenation (HHb) profiles at rest and during exercise (**A**). Changes in mean HHb with SD obtained at rest, at each stage (S1, S2, S3), and at exercise exhaustion (Peak) after 5 days of tart cherry (TC) and placebo (PL) supplementation (**B**). Suppl., supplementation; S1–S3, Stages 1–3.

**Figure 4 nutrients-15-00388-f004:**
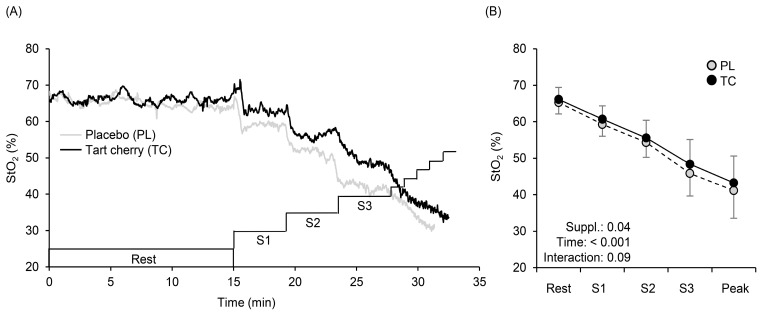
The typical time course changes of tissue oxygen saturation (StO_2_) profiles at rest and during exercise (**A**). Changes in mean StO_2_ with SD obtained at rest, at each stage (S1, S2, S3), and at exercise exhaustion (Peak) after 5 days of tart cherry (TC) and placebo (PL) supplementation (**B**).

**Figure 5 nutrients-15-00388-f005:**
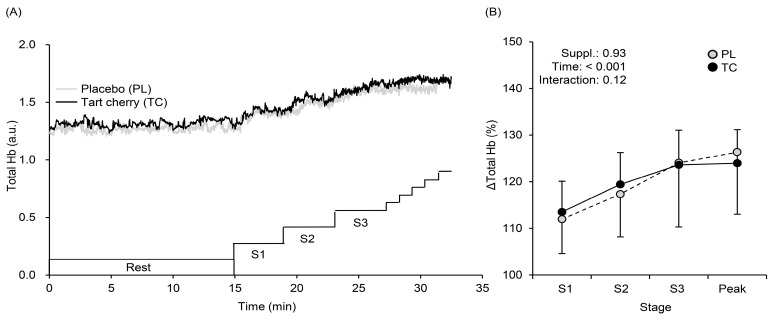
The typical time course changes in total hemoglobin (total Hb) profiles at rest and during exercise (**A**). Changes in mean total Hb with SD obtained at rest, at each stage (S1, S2, S3), and at exercise exhaustion (Peak) after 5 days of tart cherry (TC) and placebo (PL) supplementation (**B**).

**Figure 6 nutrients-15-00388-f006:**
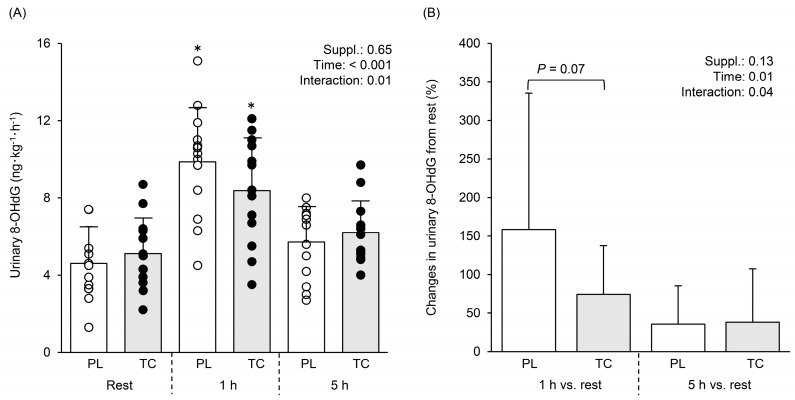
Comparisons in urinary 8-hydroxy-2deoxy guanosine (8-OHdG) excretion at rest and 1 h and 5 h post-exercise after 5 days placebo and tart cherry supplementation (**A**). Relative changes (%) from rest (**B**). White and black circles indicate individual data for PL and TC. Bar graphs indicate mean values with SD. * indicates significant difference between rest and 1 h within the same conditions.

**Table 1 nutrients-15-00388-t001:** Cardiorespiratory responses at rest, during submaximal exercise (Stages 1, 2, and 3), and at exercise exhaustion (peak) after 5 days of placebo and tart cherry supplementation.

	Placebo	Tart Cherry	Two-Way Repeated Measures ANOVA Results			
				Suppl.	Time	Interaction
$\overset{.}{V}$O_2_ (mL·min^−1^)						
Rest	271 ± 29	269 ± 32	F	1.27	327.17	1.95
Stage 1	785 ± 110	752 ± 104	*p*	0.28	<0.001	0.12
Stage 2	1175 ± 161	1153 ± 127	η^2^	0.000	0.918	0.000
Stage 3	1577 ± 151	1558 ± 167				
Peak	2058 ± 340	2080 ± 376				
$\overset{.}{V}$CO_2_ (mL·min^−1^)						
Rest	261 ± 33	261 ± 34	F	0.432	204.22	1.516
Stage 1	801 ± 126	757 ± 119	*p*	0.52	<0.001	0.21
Stage 2	1291 ± 194	1256 ± 161	η^2^	0.000	0.891	0.002
Stage 3	1813 ± 216	1691 ± 494				
Peak	2547 ± 426	2651 ± 525				
$\overset{.}{V}$_E_ (L·min^−1^)						
Rest	11.0 ± 1.8	11.3 ± 1.3	F	0.471	474.06	1.907
Stage 1	28.2 ± 3.5	27.2 ± 3.4	*p*	0.51	<0.001	0.13
Stage 2	44.6 ± 5.1	43.9 ± 4.6	η^2^	0.000	0.958	0.000
Stage 3	65.8 ± 7.2	67.0 ± 6.8				
Peak	107.9 ± 14.2	111.5 ± 13.2				
RER						
Rest	0.96 ± 0.05	0.97 ± 0.05	F	0.693	121.83	1.662
Stage 1	1.02 ± 0.06	1.01 ± 0.08	*p*	0.42	<0.001	0.17
Stage 2	1.10 ± 0.07	1.09 ± 0.06	η^2^	0.001	0.699	0.005
Stage 3	1.15 ± 0.11	1.17 ± 0.07				
Peak	1.24 ± 0.06	1.27 ± 0.07				
HR (bpm)						
Rest	80 ± 8	79 ± 8	F	0.426	608.78	0.686
Stage 1	114 ± 9	112 ± 9	*p*	0.53	<0.001	0.61
Stage 2	135 ± 10	134 ± 11	η^2^	0.000	0.926	0.000
Stage 3	156 ± 11	155 ± 9				
Peak	175 ± 11	176 ± 11				
SpO_2_ (%)						
Rest	86.6 ± 1.8	87.1 ± 1.9	F	18.53	40.10	1.75
Stage 1	82.5 ± 3.1	83.3 ± 3.3	*p*	0.001	<0.001	0.15
Stage 2	80.8 ± 3.0	81.6 ± 3.9	η^2^	0.009	0.482	0.002
Stage 3	78.2 ± 3.8	79.9 ± 4.6				
Peak	75.8 ± 5.0	76.8 ± 5.7				

Values are mean ± standard deviation. ANOVA, analysis of variance; Suppl., supplementation; $\overset{.}{V}$O_2_, pulmonary oxygen uptake; $\overset{.}{V}$CO_2_, carbon dioxide output; $\overset{.}{V}$_E_, pulmonary ventilation; RER, respiratory gas exchange ratio; HR, heart rate; bpm, beats per minute; SpO_2_, peripheral arterial oxygen saturation.

## Data Availability

Not applicable.

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
