# Peer review of "Five Days of Tart Cherry Supplementation Improves Exercise Performance in Normobaric Hypoxia"

_nutrients, 2023, doi:10.3390/nu15020388_

Round 1

Reviewer 1 Report

In the present study, Horiuchi and colleagues investigated the effects of 5 days tart cherry (TC) supplementation on exercise capacity, pulmonary gas exchange and ventilation, systemic and local tissues oxygen saturation and a urinary oxidative stress biomarker in hypoxia.  The authors report that, compared to the placebo condition, TC supplementation increased exercise capacity, local and tissue oxygen saturation and lowered post-exercise oxidative stress.  The research question is original and has the potential to make a meaningful influence on this area of research.  However, I do have some concerns that require attention to improve the manuscript.     

Title

The title is too wordy.  I would rephrase to something like "Influence of tart cherry supplementation on exercise performance in normobaric hypoxia.”

Abstract

Line 24, time not Time  

Line 27, a significant supplement x time interaction effect I assume?

Line 30, “suppressed increases in HHb and reductions in StO2” simplify and rephrase to “lower HHb and higher StO2”

The keywords needs to be populated as still in the template format.

Introduction

The authors suggest that TC may improve nitric oxide (NO) synthesis a key mechanism for its physiological and performance effects.  However, there is evidence that TC can improve performance independent of an increase in plasma [nitrite] a key NO biomarker: https://pubmed.ncbi.nlm.nih.gov/29566443/  This should be acknowledged in the introduction and discussion to provide more balance.

Lines 69-70, this sentence is unclear so should be simplified or deleted.  You are not going to be able to address spatial heterogeneity in your study as you only used 1 NIRS site. 

Line 73, “QO2-to-VO2 matching in exercising muscles” replace with stating what you will actually measure in the study (NIRS variables), not what you may be able to infer from these variables. 

Line 74, specify which tissue – skeletal muscle.

Methods

Provide some information on the fitness status of the participants.

Line 90, delete “using a lottery” here and throughout the manuscript.

Line 99, I can only see 2 experimental conditions as the familiarisation is a preliminary test not included in the data analysis. Please clarify and revise accordingly.

Lines 109, these references do not seem to fit here so remove or use appropriate references.

Line 120, you need to use the pedal range here rather than 60 rpm as you will have discontinued the test without exhaustion occurring if the first time they were cycling at 59 rpm the test was terminated. 

Figure 1, replace “on the day 5” “on day 5”

Line 126, was the composition of anthocyanins/total polyphenols of the supplement known?

The supplementation regime could be more clearly justified in the introduction.  Also, many of these studies will have used different products so cannot guarantee the same anthocyanin dose for a set mass of tart cherries, which needs to be acknowledged as a limitation in the discussion.

Lines 171-172, has this been validated?  If so, cite a paper to support this.

Results

Lines209-211, delete as from the manuscript template.

What are the mechanisms for higher SpO2 after TC supplementation?

Figure 2 legend, replace “solid lines” with “bars”?

The results could be described more clearly in section 3.3

Figures 3-5, remove “;” before the abbreviations for the conditions.

Based on the results described in section 3.5, you cannot claim with too much confidence a difference in oxidative stress between the TC and PLA conditions and this needs to be discussed appropriately in the discussion.   Essentially, tone this down slightly.

Discussion

Line 321, “exercise”

It should be acknowledged that a limitation of this study was that only 8-OHdG was measured to infer oxidative stress when it is accepted that a few biomarkers should be measured to provide a better overall picture of oxidative stress development.  Also, why was this expressed relative to body mass?  The overall discussion of the significance of your 8-OHdG results needs to be toned down.   

Lines 332-336, I do not understand your point here.  Plus, you cannot claim it is likely that TC supplementation could improve exercise performance under hypoxia independent of inhaled O2 conditions when you have not directly tested it.  Simplify and delete some of this content.

4.2, this is a poor subheading so rephrase to something link “Systemic arterial and local muscle O2 saturation”

Lines 345, subscript the 2 in O2.

What are the mechanisms for the higher SpO2 with TC supplementation?

Lines 344-346, this is repetition so I think can be deleted.

Lines 347-353, Too speculative and needs to be toned down since NIRS cannot discriminate between venules and arterioles.  Also, StO2 is spatially resolved so focus on this not the sum of its parts which can be impacted by skin blood flow.  Keep the interpretation basic and do not get too speculative.

Lines 355-358, ref 38 is older adults completing handgrip exercise so how relevant is this to your study? Perhaps provide some context here.

Briefly provide some discussion on the mechanisms by which TC may lower oxidative stress. 

In addition to 4.4, provide some potential implications of the findings.  For example, improved training quality at altitude or performance at altitude.  

Lines 411-412 this should be toned down as not supported by your post hoc analysis.

Author Response

RESPONSE TO THE REVIEWER #1

Overall comment: In the present study, Horiuchi and colleagues investigated the effects of 5 days tart cherry (TC) supplementation on exercise capacity, pulmonary gas exchange and ventilation, systemic and local tissues oxygen saturation and a urinary oxidative stress biomarker in hypoxia. The authors report that, compared to the placebo condition, TC supplementation increased exercise capacity, local and tissue oxygen saturation and lowered post-exercise oxidative stress. The research question is original and has the potential to make a meaningful influence on this area of research. However, I do have some concerns that require attention to improve the manuscript.

Response: We appreciate for your detailed reviewing. We responded point-by-point to your comments below. Edits to the manuscript are highlighted in red.

Comment: Title

The title is too wordy. I would rephrase to something like "Influence of tart cherry supplementation on exercise performance in normobaric hypoxia.”

Response: Thank you for your suggestion. We have edited the title to incorporate your and reviewer 2’s suggestion.

Line 2-3

Five-days of tart cherry supplementation improves exercise performance in normobaric hypoxia

Comment: Abstract

Line 24, time not Time 

Response: The text has been edited.

Comment: Line 27, a significant supplement x time interaction effect I assume?

Response: This has been added as you suggest.

Line 28

Moreover, a significant interaction (supplements × time) in urine

Comment: Line 30, “suppressed increases in HHb and reductions in StO2” simplify and rephrase to “lower HHb and higher StO2”

Response: We have edited as you suggest.

Line 31

due to lower HHb and higher StO2

Comment: The keywords needs to be populated as still in the template format.

Response: We added key words.

Line 33-34

Keywords: antioxidant; blood flow; DNA damage; O2 extraction; oxidative stress;

tissue oxygenation; vasodilation

Comment: Introduction

The authors suggest that TC may improve nitric oxide (NO) synthesis a key mechanism for its physiological and performance effects. However, there is evidence that TC can improve performance independent of an increase in plasma [nitrite] a key NO biomarker: https://pubmed.ncbi.nlm.nih.gov/29566443/ This should be acknowledged in the introduction and discussion to provide more balance.

Response: Thank you for introducing this useful citation. We have modified the introduction and discussion to include this citation.

Line 65-67

TC supplementation may also improve exercise performance via an antioxidant effect that prolongs the optimal cellular redox state for force production [25]. 

Comment: Lines 69-70, this sentence is unclear so should be simplified or deleted. You are not going to be able to address spatial heterogeneity in your study as you only used 1 NIRS site.

Response: We have deleted as you suggest.

Comment: Line 73, “QO2-to-VO2 matching in exercising muscles” replace with stating what you will actually measure in the study (NIRS variables), not what you may be able to infer from these variables.

Response: We have edited using NIRS metrics as you suggest.

Line 69-70

and increase tissue oxygenation in exercising muscles compared to placebo.

Comment: Line 74, specify which tissue – skeletal muscle.

Response: We specified it.

Line 71

vastus lateralis muscles

Comment: Methods

Provide some information on the fitness status of the participants.

Response: We added fitness status of participants.

Line 81

Fifteen healthy young recreationally-active volunteers

Comment: Line 90, delete “using a lottery” here and throughout the manuscript.

Response: This phrase has been deleted throughout the manuscript.

Comment: Line 99, I can only see 2 experimental conditions as the familiarization is a preliminary test not included in the data analysis. Please clarify and revise accordingly.

Response: Thank you for identifying the confusion. It has been edited to indicate there are 2 main experimental trials.

Line 97

with two experimental conditions,

Comment: Lines 109, these references do not seem to fit here so remove or use appropriate references.

Response: There have been deleted as you suggested.

Comment: Line 120, you need to use the pedal range here rather than 60 rpm as you will have discontinued the test without exhaustion occurring if the first time they were cycling at 59 rpm the test was terminated.

Response: This has been rephrased to make clearer top the reader how the workload was maintained and what criteria were used to determine exhaustion and termination of the test.

Line 118-120

no longer maintain the pedaling rate above 50 rpm despite strong verbal encouragement. The test was terminated when participants met at least one of these 3 criteria and could not maintain the pedaling rate of 50 rpm despite strong verbal encouragement [28].

Comment: Figure 1, replace “on the day 5” “on day 5”

Response: We replaced it. Please see the new figure 1. Note that in all figures, correction parts were represented as black fonts (not red) because the type of figures are not text, but, TIFF file.

Comment: Line 126, was the composition of anthocyanins/total polyphenols of the supplement known?

Comment: The supplementation regime could be more clearly justified in the introduction.  Also, many of these studies will have used different products so cannot guarantee the same anthocyanin dose for a set mass of tart cherries, which needs to be acknowledged as a limitation in the discussion.

Response: Thank you for this question. We have reworded section 2.3 and added the content to the abstract. Further, owing to different product in previous studies, we added some limitations in the methodological considerations.

Line 21-22

(ii) hypoxia with TC (200 mg anthocyanin per day for 4 days and 100 mg on day 5).

Line 124-131

Participants were assigned in a double-blinded and randomized manner to ingest TC (Tart cherry 1200 mg capsule containing 100 mg of anthocyanin, Nature’s Life, CA, USA) or a flour placebo. The TC and PL supplementations were visually indistinguishable as they were ground to powder and encapsulated in a gelatin capsule. Participants were instructed to ingest one capsule at 0800 h and one capsule at 1800 h for 4 consecutive days before the main experiment, and one capsule 2 h before exercise on the day of the main experiment. The selected TC dose was based on a recent meta-analysis [3] where previous studies daily anthocyanin supplementation ranged from 40 to 270 mg per day.

Comment: Lines 171-172, has this been validated? If so, cite a paper to support this.

Response: As written in the first draft, the measurement depth of the NIRS signal was approximately half the distance between the two fiber optic bundles according to a previous study (Patterson et al., 1989, reference number #). Although we could only estimate the depth, we edited it, and cited the citation above mentioned.

Line 160-163

This would have allowed the appropriate depth to sample from the vastus lateralis muscles, because the sum of adipose tissue and muscle thickness in vastus lateralis muscle was >20 mm [29]. Indeed, the measured adipose tissue and muscle sickness were 5.4±1.1 mm and 20.6±2.6 mm.

Comment: Results

Lines209-211, delete as from the manuscript template.

Response: Thank you for identifying this. We have deleted.

Comment: What are the mechanisms for higher SpO2 after TC supplementation?

Response: See below our response written in the discussion as the same comment seems to appear.  

Comment: Figure 2 legend, replace “solid lines” with “bars”?

Response: We apologize it again. We changed.

Comment: The results could be described more clearly in section 3.3

Response: We deleted additional results, and condensed this section as below.

Line 241-248

Figure 3 shows a typical example (panel A) and the averaged values of % changes (panel B) in the HHb. During exercise, HHb at stages 1 (40 or 30 W for men or women), 2 (80 or 60 W), and 3 (120 or 90 W) were lower with TC than PL (Figure 3A). A typical example (panel A) and averaged values of % changed (panel B) in StO2 are shown in Figure 4. During exercise, StO2 at stages 1, 2, and 3 were higher with TC than PL (Figure 4A and 4B). Total Hb gradually increased (F=39.01, η2=0.228, P<0.001) during exercise with no condition (F=0.008, η2=0.000, P=0.93) and interaction effects (F=2.06, η2=0.007, P=0.12) (Figure 5).

Comment: Figures 3-5, remove “;” before the abbreviations for the conditions.

Response: We deleted “;” from Figure 3 to 5.

Comment: Based on the results described in section 3.5, you cannot claim with too much confidence a difference in oxidative stress between the TC and PLA conditions and this needs to be discussed appropriately in the discussion. Essentially, tone this down slightly.

Response: We agreed with your comment. We further analyzed percent change from rest to 1 h and 5 h. We created new Figure 6 with a trend difference. Also, we added some descriptions in the results and discussion.

Line 289-291

Tukey post-hoc tests revealed that 1 h post-exercise 8-OHdG was increased from rest after PL and TC (Figure 6A, P<0.05) with the increase tending to be greater after PL than TC (Figure 6B, P=0.07).

Line 353-358

The present study identified an attenuated increase in the oxidative stress marker urinary 8-OHdG excretion. Indeed, 1 h post exercise the increase from rest in urinary 8-OHdG was halved after TC compared to PL (160% PL vs. and 75% TC Figure 6). These findings are agreed with previous studies that have also reported that TC supplementation attenuated an increase in lipid peroxide as an oxidative stress marker [5,7] and increased total antioxidant status [5].

Comment: Discussion

Line 321, “exercise”

Response: We edited it.

Comment: It should be acknowledged that a limitation of this study was that only 8-OHdG was measured to infer oxidative stress when it is accepted that a few biomarkers should be measured to provide a better overall picture of oxidative stress development. Also, why was this expressed relative to body mass? The overall discussion of the significance of your 8-OHdG results needs to be toned down.   

Response: At first, we added some considerations in the discussion. Also, presentation per body weight per given time is quite common for this index because it has been considered that urinary 8-OHdG excretion would be proportional to body mass. In the present study, same participants performed two experimental conditions. In this regard, dividing by body mass may not be needed. However, we sampled urine at 3 time points, and thus, to obtain more precise data (e.g., to eliminate ~200 g body weight change effects after exercise etc), we used the unit as ng/kg/h. For confirmation, we cited new citations that represented urinary 8-OHfG level as per body mass per given time

Line 186-187

Based on a previous study [33], urinary 8-OHdG excretion was calculated and represented per individual body weight per given time, i.e., as the unit of “ngï½¥kg-1ï½¥h-1”. 

Line 379-383

Third, rather than assessing a broad range of oxidative stress markers we evaluated only one oxidative stress marker urinary 8-OHdG excretion. However, the urinary excretion of 8-OHdG has been recognized as a stable biomarker of DNA oxidative damage and reflects overall systemic oxidative stress level [55].

Comment: Lines 332-336, I do not understand your point here. Plus, you cannot claim it is likely that TC supplementation could improve exercise performance under hypoxia independent of inhaled O2 conditions when you have not directly tested it. Simplify and delete some of this content.

Response: We completely rewrote this sentence. Please see the main text.

Line 305-311

4.1. Exercise performance in normobaric hypoxia

Previous studies have shown TC supplementation improves exercise performance in normoxia [1,4]. The current study extends these findings by showing for the first time that TC supplementation can also improve exercise performance in a hypoxic environment. Comparing the effect sizes of this (d = 0.80) and a previous study (d = 0.78) [1] highlights the improvement in exercise performance after TC supplementation is similar irrespective of different inspired oxygen concentrations.

Comment: 4.2, this is a poor subheading so rephrase to something link “Systemic arterial and local muscle O2 saturation”

Response: As you suggested, we changed the title off subheading (Line 333).

Comment: Lines 345, subscript the 2 in O2.

Response: We corrected this.

Comment: What are the mechanisms for the higher SpO2 with TC supplementation?

Response: We added some potential explanations.

Line 321-331

StO2 at any muscles is calculated by dividing HbO2 by total Hb (HbO2 plus HHb). In the present study, total Hb gradually increased with no differences between the conditions (Figure 5), which was associated with no differences in microvascular Hb volume [37-39]. Therefore, higher StO2 with TC might be caused either by higher HbO2 and/or lower HHb. Notably, HbO2 is known to be affected by cutaneous blood flow [40], and thus, we cannot completely exclude HbO2 effects on cutaneous circulation. However, we also found lower HHb after TC supplementation (Figure 3), indicating sufficient QO2 and TC-induced vasodilation in the vastus lateralis (exercising) muscle. Therefore, this may lead to higher StO2 and SpO2 which may be more efficient metabo-lism during exercise. We must acknowledge this is speculative but, this hypothesis has been suggested using a vasodilatory supplement in a previous study [41].

Comment: Lines 344-346, this is repetition so I think can be deleted.

Response: We arranged this sentence.

Line 316-320

Lower HHb suggests enhancements in QO2-to-VO2 matching, resulting in lower fractional O2 extraction during incremental exercise to meet muscle demands, especially in the vastus lateralis muscle [35,36], perhaps, due to greater venous O2 content, as slightly higher arterial O2 content (SpO2 as an index of arterial O2 content) was observed with TC (Table 1).

Comment: Lines 347-353, Too speculative and needs to be toned down since NIRS cannot discriminate between venules and arterioles. Also, StO2 is spatially resolved so focus on this not the sum of its parts which can be impacted by skin blood flow. Keep the interpretation basic and do not get too speculative.

Response: We edited and toned down as below.

Line 321-331

StO2 at any muscles is calculated by dividing HbO2 by total Hb (HbO2 plus HHb). In the present study, total Hb gradually increased with no differences between the conditions (Figure 5), which was associated with no differences in microvascular Hb volume [37-39]. Therefore, higher StO2 with TC might be caused either by higher HbO2 and/or lower HHb. Notably, HbO2 is known to be affected by cutaneous blood flow [40], and thus, we cannot completely exclude HbO2 effects on cutaneous circulation. However, we also found lower HHb after TC supplementation (Figure 3), indicating sufficient QO2 and TC-induced vasodilation in the vastus lateralis (exercising) muscle. Therefore, this may lead to higher StO2 and SpO2 which may be more efficient metabo-lism during exercise. We must acknowledge this is speculative but, this hypothesis has been suggested using a vasodilatory supplement in a previous study [41].

Comment: Lines 355-358, ref 38 is older adults completing handgrip exercise so how relevant is this to your study? Perhaps provide some context here.

Response: Yes, we agree with your comment. We provided one context here.

Line 336-339

However, this study was conducted using a local muscle exercise (handgrip) for older adults [43]. Therefore, further investigation is warranted to confirm this underlying mechanism in exercise involving large muscle groups such a s completed in the present study.

Comment: Briefly provide some discussion on the mechanisms by which TC may lower oxidative stress.

Response: We added a following possible mechanism.

Line 364-367

More detail, it has been suggested that anthocyanins activate Nrf-2 which increase ex-pressions of detoxifying enzymes and antioxidant enzymes, resulting in eliminate re-active oxygen spices and oxidant-induced injury cells [52,53]. Our study design cannot clarify these possible mechanisms, and hence, future studies are needed.

Comment: In addition to 4.4, provide some potential implications of the findings. For example, improved training quality at altitude or performance at altitude. 

Response: We added as below.

Line 388-392

For a practical implication, examples include mountain stages in cycling (up to 2,800 m in Tour de France) and mountain running events (e.g., at Mount Fuji, up to 3,776 m in Japan, or at Pikes Peak, up to 4,300 m in Colorado, USA). Thus, our findings potentially can be applied to these elite sports. 

Comment: Lines 411-412 this should be toned down as not supported by your post hoc analysis.

Response: We toned down.

Line 397-399

Moreover, TC supplementation may attenuate oxidative stress, as indicated by attenu-ated urinary 8-OHdG excretion rate 1 h post-exercise.

Other response: Based on both reviewer’s suggestions, we added newly 10 references. Accordingly, numbering has been changed. Also, our manuscript has been edited by native English speaker again, and thus, some terms and descriptions have been edited.

Reviewer 2 Report

Comments to the Author:

I thank to the editors for the opportunity to review this study, beside I would also like to congratulate the authors for the made effort in their study. The present manuscript by Horiuchi et al., analyzed “Application of tart cherry supplement as an antioxidant on energetic adjustment and on exercise performance under hypoxia”. The authors investigated the effects of tart cherry supplementation on hypoxic exercise performance. The main problem with this study is that the authors say it was a randomized, double-blind, placebo-controlled, placebo-controlled crossover study. However, it lacks the normoxia situation with and without supplementation. It can be said that it is an incomplete study. Therefore, tart cherry consumption may be beneficial in hypoxia but may also be beneficial in normoxia. But this we cannot know. Besides, currently the paper needs a lot of information and clarification of many doubts in the experimental design. Furthermore, there are several issues that need to be resolved.

1.      The authors argue in the introduction that there is controversy as to whether or not tart cherry consumption could improve sports performance at sea level. However, the authors then go on to discuss hypoxia. Therefore, there is no clear line of research. That is, we clarify whether the consumption of tart cherry definitely improves sports performance at sea level or whether the intake of this product is beneficial in physical activity in hypoxia. You must generate a coherent introduction to what you are going to investigate. Besides, the authors should add more information that clearly justifies the use of tart cherry to alleviate excessive oxidative stress during physical activity in a hypoxic situation.

2.      Where the authors say, "A previous study reported that 7 days of TC supplementation improved time-trial performance by ~4.7% (effect size = 0.78)." the citation of the study should be added.

3.      I am glad that the authors have used the G*Power program to perform the sample power calculation. For that reason, they should add at least a reference of that program, it is one of the rules argued by the author who designed the program.

4.      The authors should specify which type of hypoxia is high or moderate hypoxia. This should be made clear in the main document and in the title.

5.      The authors used 4 women for their study, and given that women can have more or less oxidative stress depending on the ovarian cycle they are in, how have you managed this problem?

6.      It is not clear from the methods section how many days the participants were taking the CT supplementation. Indeed, the authors argue that with 7 days other authors achieved improvements in performance. However, it is not made clear how many days were chosen for this study. On the other hand, it should be made clear that the consumption of this supplement was chronic and not acute. This issue should be made clear both in the main paper and in the title.

7.      Could the authors add any references on the guidelines taken to perform the maximal incremental leg cycling test? I would also like to know what the justification was for using this type of test and not another, such as through running or increasing the incline of the treadmill.

8.      I have a question for the authors. Are there any side effects with the consumption of this type of supplement, or have there been any undesirable effects in the participants? Indeed, two participants dropped out of the study because of problems.

9.      The authors state "TC supplementation improved the time to exhaustion compared with that of PL (940 ± 84 s with TC vs. 912 ± 63 s with PC, t = 2.87, Cohen’s d = 0.80, P = 0.01) (Figure 2). Specifically, 9 of 13 participants reported improved exercise performance", but where was it obtained in normoxia or hypoxia?

10.  The authors need to make more effort in the discussion section and give a better justification for the improved performance under hypoxia. As far as the oxygen and systemic level variables are focused on repeating the data from the results section, please focus on discussing the data.

Author Response

RESPONSE TO THE REVIEWER #2

Overall comment: I thank to the editors for the opportunity to review this study, beside I would also like to congratulate the authors for the made effort in their study. The present manuscript by Horiuchi et al., analyzed “Application of tart cherry supplement as an antioxidant on energetic adjustment and on exercise performance under hypoxia”. The authors investigated the effects of tart cherry supplementation on hypoxic exercise performance. The main problem with this study is that the authors say it was a randomized, double-blind, placebo-controlled, placebo-controlled crossover study. However, it lacks the normoxia situation with and without supplementation. It can be said that it is an incomplete study. Therefore, tart cherry consumption may be beneficial in hypoxia but may also be beneficial in normoxia. But this we cannot know. Besides, currently the paper needs a lot of information and clarification of many doubts in the experimental design. Furthermore, there are several issues that need to be resolved.

Response: We appreciate for your detailed reviewing. We responded point-by-point to your comments below. Edits to the manuscript are highlighted in red.

Comment 1: The authors argue in the introduction that there is controversy as to whether or not tart cherry consumption could improve sports performance at sea level. However, the authors then go on to discuss hypoxia. Therefore, there is no clear line of research. That is, we clarify whether the consumption of tart cherry definitely improves sports performance at sea level or whether the intake of this product is beneficial in physical activity in hypoxia. You must generate a coherent introduction to what you are going to investigate. Besides, the authors should add more information that clearly justifies the use of tart cherry to alleviate excessive oxidative stress during physical activity in a hypoxic situation.

Response 1: We modified introduction. See below the edited parts.

Line 48-55

It should be noted that these studies were conducted only under normoxia [1,4-6]. An-other distinctive advantage of TC supplementation has been proposed to reduce subset of oxidative stress markers [7,8]. However, whether this property would improve exercise performance seems to be uncertain.

Hypoxic exposure elevates several biomarkers of oxidative stress such as 8-hydro-2’ deoxyguanosine (8-OHdG) [9-13] or 2-thiobarbituric acid reactive substances [14-16] as indexes of oxidative damage to DNA or lipids, as well as exercise-induced elevated oxidative stress [17].

Comment 2: Where the authors say, "A previous study reported that 7 days of TC supplementation improved time-trial performance by ~4.7% (effect size = 0.78)." the citation of the study should be added.

Response 2: We added it.

Comment 3: I am glad that the authors have used the G*Power program to perform the sample power calculation. For that reason, they should add at least a reference of that program, it is one of the rules argued by the author who designed the program.

Response 3: We added the reference.

Comment 4: The authors should specify which type of hypoxia is high or moderate hypoxia. This should be made clear in the main document and in the title.

Response 4: We edited the title using normobaric hypoxia and specified clearly for readers. To the best of our knowledge, no gold standard to discriminate the threshold either high or moderate hypoxia. Moreover, this discrimination is not a main issue in the present study.

TITLE: Five-days of tart cherry supplementation improves exercise performance in normobaric hypoxia

Comment 5: The authors used 4 women for their study, and given that women can have more or less oxidative stress depending on the ovarian cycle they are in, how have you managed this problem?

Response 5: We added it in the method section, alternatively, we deleted this sex difference effects from the methodological considerations

Line 88-90

Women were studied during the follicular phase just after menstruation based on their basal body temperature and self-report [27].

Comment 6: It is not clear from the methods section how many days the participants were taking the CT supplementation. Indeed, the authors argue that with 7 days other authors achieved improvements in performance. However, it is not made clear how many days were chosen for this study. On the other hand, it should be made clear that the consumption of this supplement was chronic and not acute. This issue should be made clear both in the main paper and in the title.

Response 6: We edited methodology and the title (see also above).

Line 124-131

Participants were assigned in a double-blinded and randomized manner to ingest TC (Tart cherry 1200 mg capsule containing 100 mg of anthocyanin, Nature’s Life, CA, USA) or a flour placebo. The TC and PL supplementations were visually indistin-guishable as they were ground to powder and encapsulated in a gelatin capsule. Par-ticipants were instructed to ingest one capsule at 0800 h and one capsule at 1800 h for 4 consecutive days before the main experiment, and one capsule 2 h before exercise on the day of the main experiment. The selected TC dose was based on a recent me-ta-analysis [3] where previous studies daily anthocyanin supplementation ranged from 40 to 270 mg per day.

Comment 7: Could the authors add any references on the guidelines taken to perform the maximal incremental leg cycling test? I would also like to know what the justification was for using this type of test and not another, such as through running or increasing the incline of the treadmill.

Response 7: Incremental exercise test type is varied, and no gold standard. Compared to running, leg cycling has one advantage to obtain NIRS signals without fluctuation. But, we considered that it is not needed to mention in the main text.

Comment 8: I have a question for the authors. Are there any side effects with the consumption of this type of supplement, or have there been any undesirable effects in the participants? Indeed, two participants dropped out of the study because of problems.

Response 8: Not sure. Indeed, one participant clamed with PL, but the other claimed with TC. If all participants claimed only with PL, it can be said that the reason for dropped out may depend on hypoxic condition, not on TC condition. But, we cannot know as it may also depend on individual physical condition level on that day.

Line 202-203

one participant claimed under PL condition, and the other claimed under TC condition

Comment 9: The authors state "TC supplementation improved the time to exhaustion compared with that of PL (940 ± 84 s with TC vs. 912 ± 63 s with PC, t = 2.87, Cohen’s d = 0.80, P = 0.01) (Figure 2). Specifically, 9 of 13 participants reported improved exercise performance", but where was it obtained in normoxia or hypoxia?

Response 9: We edited the subheading for readers to follow it.

Comment 10: The authors need to make more effort in the discussion section and give a better justification for the improved performance under hypoxia. As far as the oxygen and systemic level variables are focused on repeating the data from the results section, please focus on discussing the data.

Response 10: We edited half of the discussion. Please see the main text.

Other response: Based on both reviewer’s suggestions, we added newly 10 references. Accordingly, numbering has been changed. Also, our manuscript has been edited by native English speaker again, and thus, some terms and descriptions have been edited.

Round 2

Reviewer 2 Report

For the author:

I appreciate authors’ effort. The authors have obviously spent considerable time revising the manuscript and their hard work is clearly paying off. This manuscript is drastically improved from the original submission. The message is very clear, the language is much more clean, and the issues in the first version were corrected. Besides, the authors have answered all my comments successfully. For this reason, I encourage to editor to consider this manuscript for publication for the interesting value of the study realized, that now it is a much more robust study.